# Prevalence of Orthosomnia in a General Population Sample: A Cross-Sectional Study

**DOI:** 10.3390/brainsci14111123

**Published:** 2024-11-06

**Authors:** Haitham Jahrami, Khaled Trabelsi, Waqar Husain, Achraf Ammar, Ahmed S. BaHammam, Seithikurippu R. Pandi-Perumal, Zahra Saif, Michael V. Vitiello

**Affiliations:** 1Government Hospitals, Manama 329, Bahrain; zahra-saif@outlook.com; 2Department of Psychiatry, College of Medicine and Medical Sciences, Arabian Gulf University, Manama 329, Bahrain; 3High Institute of Sport and Physical Education of Sfax, University of Sfax, Sfax 3000, Tunisia; trabelsikhaled@gmail.com; 4Research Laboratory: Education, Motricity, Sport and Health, EM2S, LR19JS01, University of Sfax, Sfax 3000, Tunisia; 5Department of Humanities, COMSATS University Islamabad, Islamabad Campus, Park Road, Islamabad 45550, Pakistan; drsukoon@gmail.com; 6Department of Training and Movement Science, Institute of Sport Science, Johannes Gutenberg-University Mainz, 55099 Mainz, Germany; ammar.achraf@ymail.com; 7Research Laboratory, Molecular Bases of Human Pathology, LR19ES13, Faculty of Medicine of Sfax, University of Sfax, Sfax 3000, Tunisia; 8Department of Medicine, University Sleep Disorders Center and Pulmonary Service, King Saud University, Riyadh 11324, Saudi Arabia; ashammam2@gmail.com; 9The Strategic Technologies Program of the National Plan for Sciences and Technology and Innovation in the Kingdom of Saudi Arabia, Riyadh 11324, Saudi Arabia; 10Centre for Research and Development, Chandigarh University, Mohali 140413, Punjab, India; pandiperumal2023@gmail.com; 11Division of Research and Development, Lovely Professional University, Phagwara 144411, Punjab, India; 12Department of Psychiatry & Behavioral Sciences, University of Washington, Seattle, WA 98195-6560, USA; vitiello@uw.edu

**Keywords:** anxiety, insomnia, orthosomnia, sleep tracking, wearable devices

## Abstract

Background/Objectives: Orthosomnia has become a concern in the field of sleep medicine. The purpose of this cross-sectional study was to estimate the prevalence of orthosomnia in the general population. Methods: We collected data from 523 participants via the Generalized Anxiety Disorder Scale, Anxiety and Preoccupation about Sleep Questionnaire, and Athens Insomnia Scale. Additionally, we gathered information about participants’ use of commercial sleep-tracking wearable devices. Results: We developed a four-criteria algorithm to identify cases of orthosomnia: ownership of a wearable sleep-tracking device, AIS score ≥ 6, GAD-7 score ≤ 14, and APSQ score ≥ 40 or APSQ score ≥ 35 or APSQ score ≥ 30, for conservative, moderate, and lenient prevalence estimates, respectively. One hundred seventy-six (35.8%) (95% CI 34.6–40.1%) participants regularly used sleep-tracking devices. The prevalence rates of algorithm-identified orthosomnia in the study sample were: 16 participants (3.0%, 95% CI 1.6–4.5%), 45 participants (8.6%, 95% CI 6.2–11.0%), 73 participants (14.0%, 95% CI 10.9–16.9%) for the for conservative, moderate, and lenient prevalence estimates, respectively. Individuals with orthosomnia were not significantly different in terms of age and sex. The cases consistently had higher AIS scores than non-cases across all APSQ cutoffs, indicating more severe insomnia symptoms, with significant differences observed at each cutoff point. Conclusions: This study offers initial insights into the prevalence of orthosomnia within our sample at a specific time. The findings reveal notable rates of orthosomnia among individuals using sleep-tracking devices; however, we must acknowledge the limitations inherent in a cross-sectional design.

## 1. Introduction

Optimal sleep is essential for our physical and mental health, well-being, and good quality of life, but a restful night’s sleep is a goal that many individuals find difficult to attain [1]. Interest in monitoring sleep and enhancing sleep quality has increased recently as sleep-tracking devices are becoming more accessible and popular worldwide [2]. This trend has seen the emergence of a new phenomenon called “orthosomnia”, which was coined by Baron et al. in 2017 [3]. Orthosomnia refers to an unhealthy preoccupation with achieving perfect sleep as defined by these tracking devices [3,4].

Although case studies and anecdotal reports have shed light on the potential negative impact of orthosomnia on sleep quality and daytime functioning, previous studies have not attempted to estimate its prevalence in the general population [3,5]. This lack of knowledge hinders our understanding of the extent of the problem and its potential public health implications [6]. On the basis of our clinical observations, we hypothesized that the prevalence of orthosomnia in the general population is approximately 3–5% [6,7,8]. The purpose of this study was to address this gap by quantitatively estimating the prevalence of orthosomnia in a general population sample. To do so, we combined validated scales for anxiety and sleep-related concerns with information about wearable device usage.

As a secondary objective, we examined age and sex as potential risk factors associated with orthosomnia. Understanding such demographic characteristics as risk factors could provide valuable insights for targeted prevention and intervention strategies.

Study findings may contribute to the growing body of knowledge on the intersection of technology and sleep health and inform healthcare providers, sleep medicine specialists, and public health officials in addressing this emerging concern.

Optimal sleep is essential for physical and mental health, well-being, and a good quality of life [1]. However, many individuals struggle to attain a restful night’s sleep [9]. In recent years, interest in monitoring and enhancing sleep quality has surged, partly driven by the increasing accessibility and popularity of sleep-tracking devices worldwide [2]. This trend has given rise to a new phenomenon called “orthosomnia”, a term coined by Baron et al. in 2017 [3]. Orthosomnia refers to an unhealthy preoccupation with achieving perfect sleep as defined by these tracking devices [3,4].

The growing prevalence of sleep-tracking technology has revolutionized how individuals engage with their sleep patterns [10]. Wearable devices and smartphone applications now offer users detailed metrics on sleep duration, quality, and cycles [10]. While these tools provide valuable insights, they may also heighten anxiety and obsession over sleep data for some users [3,4]. This preoccupation can paradoxically lead to sleep disturbances, creating a cycle of anxiety and poor sleep [5].

Recent research has begun to explore the implications of widespread sleep tracking [2]. Robbins et al. conducted a systematic review of studies using commercial sleep-tracking technology, highlighting both the potential benefits and limitations of these devices [2]. Robbins’ team found that while sleep trackers can increase user engagement with sleep health, concerns exist about their accuracy and the potential for increased anxiety in some users [2]. Glazer Baron et al. further examined how consumer sleep technology data are utilized in behavioral sleep medicine interventions, underscoring the growing integration of these technologies in clinical practice [4].

The concept of orthosomnia has gained increasing attention in sleep medicine literature. Baron et al.’s original case series [3] described patients whose sleep issues worsened due to strict adherence to sleep tracker data, which often conflicted with objective sleep measures [3]. This phenomenon raises important questions about the psychological impact of quantifying sleep and the potential for technology-induced sleep anxiety [3].

Despite growing clinical awareness of orthosomnia, in sleep medicine we still lack a clear understanding of its prevalence and characteristics in the general population [6]. Although case studies and anecdotal reports have highlighted the negative impact of orthosomnia on sleep quality and daytime functioning [3,5], previous studies have not estimated its prevalence in the general population [6]. This absence of epidemiological data limits our understanding of the issue’s extent and its potential public health implications [6,8].

Moreover, the relationship between orthosomnia and demographic factors such as age and sex remains unexplored [6]. Identifying these potential risk factors could provide valuable insights for targeted prevention and intervention strategies. Moreover, we need to investigate the association between orthosomnia and insomnia symptoms, as this may inform differential diagnosis and treatment approaches in sleep medicine.

The present cross-sectional study aims to address these knowledge gaps by quantitatively estimating the prevalence of orthosomnia in a general population sample. We developed and applied a novel classification algorithm that combines several validated scales for insomnia, anxiety, and sleep-related concerns with information about wearable device usage. As secondary objectives, we examined age and sex as potential risk factors associated with orthosomnia and explored the relationship between orthosomnia and insomnia symptoms. On the basis of our clinical observations, we hypothesized that the prevalence of orthosomnia in our sample would be approximately 3–5% [6,7,8].

Our findings may contribute to the growing body of knowledge on the intersection of technology and sleep health. By providing the first quantitative estimates of orthosomnia prevalence and associated factors, we aim to inform healthcare providers, sleep medicine specialists, and public health officials about this emerging concern. Furthermore, this research may guide future investigations into the long-term impacts of sleep tracking on sleep perception and behavior, ultimately leading to more effective strategies for promoting healthy sleep in the digital age.

## 2. Materials and Methods

### 2.1. Participants and Procedure

We conducted a cross-sectional survey of 523 adults (aged 18–70) who were recruited through online platforms and local community advertisements. All participants were required to have no chronic medical or psychiatric conditions, as well as no significant health issues in the four weeks prior to the study which required attention from healthcare providers and/or prescribed pharmacological treatment. We ensured that no incentives were offered for participation, thus preserving the integrity of voluntary participation and informed consent. The study protocol and data collected in that original study have been previously published [11]. The current study is a secondary analysis of those data [11].

### 2.2. Sample and Sample Size

The study sample included 523 adults aged 18–70 years, with a median = 21 years, IQR = 5. The majority of the participants were female (81%). The majority of participants were single (83%, n = 435), and 17% (n = 88) were married.

To justify the sample size, we based our calculations on the assumption of a 3–5% prevalence rate for orthosomnia, which is a condition with no known estimates. The 3–5% prevalence rate was selected to ensure statistical reliability given that in the adult population approximately 25% of individuals own a sleep tracker [2,6] and approximately 20% experience insomnia symptoms [12]. A minimum sample size of 385 participants was required to achieve a 95% confidence level with a 5% margin of error for detecting such a rare prevalence. Our available general population sample of 523 participants well exceeded this estimated minimum requirement, ensuring sufficient statistical power to detect the prevalence of orthosomnia and conduct meaningful analyses.

### 2.3. Measures

Participants completed an online questionnaire that included several validated scales. The 7-item Generalized Anxiety Disorder Scale (GAD-7) measures anxiety symptoms with each item scored from 0 to 3 [13]. Total scores range from 0 to 21, with higher scores indicating greater GAD symptom severity [13]. The GAD-7 has shown excellent internal consistency (α), good test‒retest reliability (TRT), and good convergent validity (CV) with anxiety measures [13]. The values for α, TRT, and CV are 0.92, 0.83, and >0.75, respectively [13]. Using a cutoff score of 10 (indicating moderate anxiety) the GAD-7 demonstrated a high sensitivity of 89% and specificity of 82% for diagnosing generalized anxiety disorder [13].

The Anxiety and Preoccupation about Sleep Questionnaire (APSQ) is a 10-item scale that uses a 5-point Likert scale, with total scores ranging from 10–50 [14]. Higher scores on the APSQ indicate greater sleep-related anxiety and preoccupation [14]. The values for α, TRT, and CV are 0.91, 0.80, and approximately 0.50, respectively [11,14,15]. The APSQ has also shown good discriminant/divergent validity in distinguishing between individuals with and without insomnia [15].

The Athens Insomnia Scale (AIS) is an 8-item scale that measures insomnia symptoms based on ICD-10 criteria, each rated from 0 to 3 [16]. Total scores range from 0 to 24 with higher scores indicating more severe insomnia symptoms [16]. The values for α, TRT, and CV are 0.85, 0.86, and >0.90, respectively [16]. When a cutoff score of 6 was used, the AIS demonstrated a high sensitivity of 93% and specificity of 85% for diagnosing insomnia [16].

In addition to these validated scales, participants were asked if they owned and regularly used commercial wearable devices to monitor their sleep in the past 4 weeks. Although this single-item measure is not a validated scale, it provides valuable information for assessing the prevalence of sleep-tracking behavior in a sample and identifying individuals who may be at greater risk for orthosomnia.

### 2.4. Orthosomnia Classification Algorithm

The classification algorithm for orthosomnia was developed by the first author based on the literature and clinical expertise. The algorithm incorporates multiple criteria to ensure a comprehensive assessment. Specifically, the algorithm classifies participants as having orthosomnia if they meet all four prespecified conditions. First, participants must own and regularly use a sleep-tracking wearable device to track their sleep. This condition is crucial, as it represents a key trigger for the excessive monitoring and tracking of sleep, which is characteristic of orthosomnia. Second, a GAD-7 score less than 15 is needed to exclude patients with generalized anxiety disorder [13]. This condition was specifically chosen to rule out individuals with severe anxiety that might not be primarily sleep-related, such that the observed anxiety is more likely associated with sleep concerns rather than the result of a broader anxiety disorder. The regular use of such devices is often the catalyst for the development of sleep-related anxiety and preoccupation in individuals with orthosomnia. Third, an AIS score greater than 6 is necessary to indicate the presence of insomnia symptoms. This cutoff was selected because it aligns with the established threshold for identifying clinically significant insomnia symptoms on the AIS, ensuring that participants classified as having orthosomnia are experiencing sleep difficulties [16].

To determine the best threshold for the ASPQ total score to be used in the definition of orthosomnia, we tested various cutoffs ranging from 15 to 45. The APSQ score ranges from a minimum of 10 to a maximum of 50, with higher scores indicating more severe symptoms and no specified cutoff. In this research, we employed multiple cumulative cutoffs. Specifically, for an APSQ score of ≥45, the range corresponds to scores from 10 to 45; for ≥40, the range is 10 to 40; for ≥35, it includes scores from 10 to 35; for ≥30, the range is 10 to 30; for ≥25, it covers scores from 10 to 25; for ≥20, the range is 10 to 20; and finally, for ≥15, it includes scores from 10 to 15. We stopped at the cutoff of 15 because it became clear that it was saturated; specifically, the number of orthosomnia cases defined by an APSQ score of ≥20 was equal to those defined by a score of ≥15 (i.e., orthosomnia cases APSQ ≥ 20 = orthosomnia cases APSQ ≥ 15). Ultimately, an ASPQ total score >35 (immediately above sample mean) was chosen to demonstrate high levels of sleep-related obsession, anxiety, and preoccupation. This threshold was selected to capture individuals with notably high sleep anxiety, as lower scores might not reflect the intense preoccupation with sleep central to orthosomnia. Additionally, those with an ASPQ total score of >30 were considered an at-risk group and could be used to estimate a more lenient prevalence rate compared to those with a score of >35. Conversely, an ASPQ total score of >40 was considered a conservative threshold, identifying cases that meet all theoretical diagnostic criteria for orthosomnia. By combining these carefully selected criteria, the algorithm identifies individuals who exhibit significant sleep-related anxiety and preoccupation in the context of using sleep-tracking devices while experiencing insomnia symptoms but not meeting the threshold for severe GAD. A customized script in the language of R for statistical computing was created and made available on GitHub/in the Appendix A to ensure reproducibility and transparency.

### 2.5. Ethics

The research adhered to the STROBE guidelines for cross-sectional studies. This secondary analysis was reviewed and approved by the Psychiatric Hospital Research Committee (PREC-0724-03) on 25 July 2024. Informed consent was obtained from all participants during the initial data collection, previously published. The study followed the Declaration of Helsinki principles, ensuring ethical standards in all research involving human participants. Informed consent was obtained from all subjects involved in the study.

### 2.6. Statistical Analysis

We evaluated the reliability of the measures used in the present study using Cronbach’s alpha. This statistical measure assesses the internal consistency of the scales, ensuring that the items within each scale reliably measure the same construct.

The sample was described, and the prevalence rate of orthosomnia was calculated via descriptive statistics. Since the data did not follow a normal distribution, we reported the median and interquartile range (IQR) for continuous variables and frequencies and percentages for categorical variables. The prevalence of orthosomnia was calculated as a percentage with the associated 95% confidence interval (95% CI) via the appropriate Wilson method for proportions.

To illustrate the methodology our algorithm employs in identifying orthosomnia, Figure 1 presents a flowchart that delineates the number of individuals meeting each specified criterion. Our study began with 523 participants, initially categorized based on their Generalized Anxiety Disorder (GAD-7) scores. Among these participants, 417 scored 14 or lower on the GAD-7 scale. Within this subset, 137 participants used a wearable sleep tracker, from which we identified 83 individuals diagnosed with insomnia symptoms. Further stratification based on the Anxiety and Preoccupation about Sleep Questionnaire scores revealed the following: 73 participants scored 30 or higher, 45 scored 35 or higher, and 16 scored 40 or higher. We classified individuals meeting all these criteria as “cases” of orthosomnia.

We classified those who did not meet one or more of these criteria as “non-cases”.

For sensitivity analyses, we also re-defined “non-cases” as individuals who scored 14 or lower on the GAD-7 scale, owned a sleep tracker, had an AIS score of less than 6, or an APSQ score of less than 30, providing an alternative definition of non-cases.

To compare the demographic and clinical characteristics between patients with and without orthosomnia, we used chi-square tests for categorical variables and the Mann‒Whitney U test or independent samples *t*-test for the continuous variable age and AIS. We used rank biserial correlation to assess the effect size of the Mann–Whitney U test. We also conducted a chi-square test to explore the associations between orthosomnia status and the categorical variable sex. To address the sex imbalance, we repeated the chi-square test by matching for age and wearable status, comparing 101 males and 101 females to ensure a balanced and reliable analysis of the association between orthosomnia status and sex.

We also employed the generalized linear model (GLM) to analyze the relationship between orthosomnia status and predictor variables, including age and sex. Researchers chose logistic regression due to the binary nature of the dependent variable, orthosomnia cutoff at 40, and applied a logit link function. The model assumes a binomial distribution to account for the dichotomous outcome and evaluates model fit and complexity using metrics such as R-squared, AIC, and BIC.

All the statistical analyses were conducted via R version 4.4.1 (Race for Your Life), which was released on 14 June 2024. A *p*-value less than 0.05 was considered statistically significant for all the statistical tests.

## 3. Results

### 3.1. Sample Characteristics

The final sample consisted of 523 participants (median = 21 years, IQR = 5; 81% female). Among these, 187 (35.8%, 95% CI 34.6–40.1%) reported owning and regularly using a sleep-tracking wearable device. See Table 1.

The scales used in this study yielded the following Cronbach’s alpha values: the GAD-7 had a reliability coefficient of 0.89, the APSQ showed a coefficient of 0.91, and the AIS recorded a coefficient of 0.88. These values indicate that each measure possesses high internal consistency.

A Mann–Whitney U test revealed a statistically significant difference in ASPQ total scores between groups based on “using a sleep-tracking wearable device”. Results of the groups not using a sleep-tracking wearable device (median = 30, IQR = 12) and using a sleep-tracking wearable device (median = 35, IQR = 9) were U (521) = 20,581, *p* < 0.001.

### 3.2. Prevalence Rates and Characteristics of Orthosomnia

Our classification algorithm indicates that the prevalence of orthosomnia varies depending on the ASPQ score threshold. Using a conservative cutoff of ≥40, we classified 16 participants (3.0%, 95% CI 1.6–4.5%) as having orthosomnia. At a moderate cutoff of ≥35, this number increased to 45 participants (8.6%, 95% CI 6.2–11.0%), and with a lenient cutoff of ≥30 or higher, 73 participants (14.0%, 95% CI 10.9–16.9%) were identified. See Table 1. Further increases in leniency beyond the ≥30 ASPQ cutoff yielded minimal changes in classification: 76 participants (14.5%, 95% CI 11.5–17.6%) met the criteria at a cutoff of ≥25, and 82 participants (15.7%, 95% CI 12.6–18.8%) were classified at ≥20. Conversely, only six individuals (1.1%, 95% CI 0.25–2.0%) met the criteria using a cutoff of ≥45. See Table 1.

At an ASPQ cutoff of >30, the overall prevalence of orthosomnia reached 14.0% (73/523), with 14.2% (60/422) among females and 12.9% (13/101) among males. Statistical analysis using the χ^2^ test revealed no significant difference between sexes (χ^2^ = 0.123, *p* = 0.726, Cramer’s V = 0.0153), with an odds ratio of 1.12 (95% CI: 0.59, 2.13) and a difference in proportions of 0.0135 (95% CI −0.0599, 0.0868). As the ASPQ cutoff increased to >35, the overall prevalence dropped to 8.6% (45/523), with females at 9.5% (40/422) and males at 5.0% (5/101). This reduction also showed no statistical significance (χ^2^ = 2.121, *p* = 0.145, Cramer’s V = 0.0637), featuring an odds ratio of 2.01 (95% CI 0.77, 5.23) and a difference in proportions of 0.0453 (95% CI −0.00542, 0.0960). At the highest cutoff of >40, the prevalence further decreased to 3.1% (16/523), with 3.3% (14/422) of females and 2.0% (2/101) of males affected. Again, this difference was not statistically significant (χ^2^ = 0.491, *p* = 0.483, Cramer’s V = 0.0307), with an odds ratio of 1.70 (95% CI: 0.38, 7.60) and a difference in proportions of 0.0134 (95% CI −0.0187, 0.0455). See Table 2. In the matched analysis comparing 101 males and 101 females, no statistically significant results were obtained at the APSQ cutoffs of 30, 35, and 40, all *p*-values > 0.5.

Table 3 presents the results of age comparisons between case and non-case groups at different APSQ cutoffs, using a sample size of 523. For APSQ ≥ 30, the median age of cases was 20 years (IQR = 5), while non-cases had a median age of 21 years (IQR = 5). The Mann–Whitney U test showed no significant difference (U = 14,469, *p* = 0.1, rank biserial correlation = 0.119). At an APSQ cutoff of ≥35, the median age of cases and non-cases was 21 years (IQR = 5). The difference was not significant (U = 10,130, *p* = 0.516, rank biserial correlation = 0.0581). For APSQ ≥ 40, cases had a median age of 19 years (IQR = 5), whereas non-cases had a median age of 21 years (IQR = 5). The Mann–Whitney U test approached significance (U = 2937, *p* = 0.06, rank biserial correlation = 0.276). Cases tended to be younger than non-cases at APSQ ≥ 30 and APSQ ≥ 40, but not APSQ ≥ 35, and differences were not statistically significant.

When we redefined “non-cases” as individuals who scored 14 or lower on the GAD-7 scale, owned a sleep tracker, and had an AIS score of less than 6 or an APSQ score of less than 30, 35, or 40. We observed no significant differences compared to the full sample. The results from the independent samples *t*-tests indicated that for the APSQ ≥ 30 cutoff, the Mann–Whitney U statistic was 314.50 with a *p*-value of 0.480, yielding a rank biserial correlation of 0.14. For this cutoff, the median age was 20.00 for both the “No” group (n = 10) and the “Yes” group (n = 73). Similarly, at the APSQ ≥ 35 cutoff, the Mann–Whitney U statistic was 734.00 with a *p*-value of 0.266 and a rank biserial correlation of 0.14, with median ages of 20.00 for the “No” group (n = 38) and 21.00 for the “Yes” group (n = 45). Finally, for the APSQ ≥ 40 cutoff, the Mann–Whitney U statistic was 437.50, *p* = 0.254, and rank biserial correlation was 0.18, with median ages of 21.00 for the “No” group (n = 67) and 19.00 for the “Yes” group (n = 16). These analyses indicate that the redefined criteria for “non-cases” did not yield significant differences in age across the various APSQ cutoffs.

### 3.3. Insomnia in Participants with Orthosomnia Symptoms

The results of the insomnia comparison between case and non-case groups using the Athens Insomnia Scale (AIS) are presented in Table 4. For participants with an APSQ score of ≥30, the median AIS score for cases was 8 (IQR = 2), while non-cases had a median score of 4 (IQR = 4). The Mann–Whitney U test indicated a significant difference between the groups (U = 6459, *p* < 0.001), with a rank biserial correlation of 0.607. At an APSQ cutoff of ≥35, cases had a median AIS score of 8 (IQR = 3), compared to 4 (IQR = 4) for non-cases, which was also statistically significant (U = 4030, *p* < 0.001, rank biserial correlation = 0.625). For APSQ ≥ 40, cases had a median AIS score of 9 (IQR = 5), while non-cases scored 4 (IQR = 3). This difference was confirmed by the Mann–Whitney U test (U = 1400, *p* < 0.001, rank biserial correlation = 0.655). Across all APSQ cutoffs, cases consistently exhibited higher AIS scores than non-cases, indicating more severe insomnia symptoms, with significant differences observed at each cutoff point. See Table 4.

We reexamined the insomnia comparison based on the Athens Insomnia Scale (AIS) at various APSQ cutoffs, using the redefined criteria for “non-cases” mentioned above. Our analysis revealed significant differences between the case and non-case groups. For the APSQ ≥ 30 cutoff, we calculated a Mann–Whitney U statistic of 222.50 with a *p*-value of 0.043, resulting in a rank biserial correlation of 0.39. The median AIS score for the “No” group (n = 10) was 7.00, while for the “Yes” group (n = 73) it was 8.00. At the APSQ ≥ 35 cutoff, the Mann–Whitney U statistic was 572.00, with a *p*-value of 0.009 and a rank biserial correlation of 0.33. The median AIS score for the “No” group (n = 38) remained at 7.00, and for the “Yes” group (n = 45) it was also 8.00. Finally, for the APSQ ≥ 40 cutoff, we found a Mann–Whitney U statistic of 344.50, with a *p*-value of 0.025 and a rank biserial correlation of 0.36. The median AIS score for the “No” group (n = 67) was 8.00, while for the “Yes” group (n = 16) it was 9.00. These results indicate significant differences in AIS scores between the case and non-case groups at all APSQ cutoffs, highlighting the severity of insomnia symptoms in the case group.

The GLM yielded an R-squared value of 0.02, indicating that the predictors explained a modest proportion of the variance. Age had a non-significant effect (*p* = 0.142) with an odds ratio of 1.08. Similarly, sex did not significantly impact the outcome (*p* = 0.491), showing an odds ratio of 1.69. The model converged successfully; the predictors did not significantly contribute to explaining the outcome variable.

## 4. Discussion

This study compared the estimated prevalence rates and demographic and clinical characteristics of algorithmically defined individuals with those without orthosomnia in a convenience sample of the general population. Approximately 36% of the participants used wearable sleep-tracking devices. Algorithmically defined “full-blown” orthosomnia was conservatively observed in 16 participants (3.0%, 95% CI 1.6–4.5%) which matches clinical observations. No significant sex differences were observed. Although it did not reach statistical significance (*p*-value 0.058), individuals identified as being orthosomniac tended to be younger (median age of 19 years) than non-patients were (median age of 21 years). These results suggest a greater susceptibility to orthosomnia among younger individuals (teenagers), possibly due to greater use of sleep-tracking technology and increased anxiety about sleep quality. Furthermore, younger individuals often exhibit a stronger bond with their mobile phones (nomophobia) [17] and other accompanying gadgets [7]. This could offer an additional explanation for their increased susceptibility to orthosomnia.

Our findings demonstrate that cases consistently had higher AIS scores than non-cases across all APSQ cutoffs, indicating more severe insomnia symptoms. This result aligns with previous research, which has shown that the APSQ has also shown good discriminant/divergent validity in distinguishing between individuals with and without insomnia [15].

The lack of established diagnostic criteria or validated scales for orthosomnia presents a major challenge in validating any classification method for this condition. We developed our algorithm based on current literature and expert clinical judgment, incorporating elements that align with the theoretical conceptualization of orthosomnia. Although the prevalence rate meets clinical expectations, we recognize that this alone does not validate the algorithm. We did not directly compare our algorithm’s results to expert clinical judgments on a case-by-case basis, nor did we provide specific validation data. Therefore, we should view our classification algorithm as a preliminary attempt to operationalize orthosomnia for research purposes, rather than a definitive diagnostic tool. Future research should focus on developing and validating a specific scale for orthosomnia, comparing our algorithm’s results with expert clinical diagnoses, and refining the algorithm based on emerging research and expert consensus. We appreciate the reviewer for highlighting this important issue and will explicitly discuss these limitations in future revisions of the manuscript.

It is important to consider orthosomnia in the context of the overall prevalence of insomnia. Various studies worldwide have shown that insomnia affects 10–30% of the population [18]. Importantly, a portion of the insomnia cases, potentially 3–8%, might involve orthosomnia rather than traditional insomnia. While both conditions involve sleep disturbances, orthosomnia is characterized by its unique association with sleep tracking and data preoccupation. Similarities between orthosomnia and insomnia include difficulties falling asleep or staying asleep and impairments during the day [6,14]. However, the key difference lies in the obsessive focus on sleep data and the pursuit of perfection in sleep metrics in orthosomnia patients [6,8]. This distinction highlights the need for careful assessment and differentiation in clinical settings to ensure appropriate treatment approaches [3,5]. Future research should aim to enhance our understanding of the relationship between orthosomnia and insomnia, potentially leading to more targeted interventions for each condition.

Future research in this field should explore genetic markers as potential tools for identifying and understanding orthosomnia. Specifically, investigating clock gene expression and clock gene polymorphisms in both the orthosomnia and control groups could provide valuable insights. Clock genes play crucial roles in regulating circadian rhythms and sleep patterns, and variations in these genes have been associated with various sleep disorders. By collecting blood samples and conducting genetic analyses, researchers may be able to identify specific genetic markers that are more prevalent in individuals with orthosomnia. This approach could not only aid in the identification of orthosomnia but also shed light on its underlying biological mechanisms.

These findings highlight the complex connection between sleep tracking technologies and sleep-related anxiety. While these devices can provide valuable insights into sleep patterns, our results suggest that, for some users, they may contribute to or exacerbate concerns related to sleep. In this study, we excluded familial history of insomnia or anxiety as a covariate in our analyses. However, we recognize that these factors can significantly influence the development and manifestation of insomnia symptoms. Future research should evaluate familial history as a covariate to better understand its impact on insomnia and anxiety outcomes.

A significant limitation of this study is the sex imbalance in our sample, with females making up about 80% of the participants. This overrepresentation of females could skew the results and limit the generalizability of our findings to the broader population. Although our analysis did not show significant differences in orthosomnia prevalence between males and females, the small number of male participants (n = 101) compared to females (n = 422) reduces the statistical power to detect any existing differences. This imbalance may also mask potential sex-specific patterns or risk factors associated with orthosomnia. In the matched analysis comparing equal numbers of males and females, we found a statistically significant difference in orthosomnia status at the APSQ cutoff of 30. However, at the higher cutoffs of 35 and 40, there were no statistically significant differences. This suggests that sex differences may be more apparent at lower APSQ cutoff levels.

Future studies should aim for a more balanced sex representation to provide a more accurate picture of orthosomnia prevalence and characteristics across sexes. Other limitations of this study should be acknowledged. First, our classification algorithm, although it is based on literature and clinical expertise, has not been validated against clinical diagnoses of orthosomnia. Second, the cross-sectional nature of the study prevents us from drawing causal conclusions about the relationship between wearable device use and the development of orthosomnia. Finally, our sample may not fully represent the general population, which could limit the applicability of our findings.

The practical implications of this study are large, emphasizing the importance of using clinical criteria or validated scales to identify orthosomnia. These criteria/conditions should include owning a wearable device or engaging in manual sleep tracking, experiencing insomnia symptoms, being preoccupied with sleep, not having anxiety or other medical or psychiatric disorders, and having symptoms persist for at least two weeks. Implementing these criteria in clinical practice and research is vital for accurate diagnosis and differentiation from other sleep disorders. We also suggest that developing and validating specific scales for detecting orthosomnia could greatly improve our ability to identify and study this condition in different populations.

## 5. Conclusions

This study provides initial evidence regarding the prevalence of orthosomnia in a sample of the general population, suggesting that approximately 2% of individuals may be affected. These findings underscore the need for further research on orthosomnia, which should include establishing standardized diagnostic criteria, examining its long-term effects on sleep and health, and exploring potential interventions. With the increasing use of commercial sleep-tracking technologies, healthcare providers should be aware of the possibility of experiencing orthosomnia and consider screening patients who are experiencing sleep concerns. Future research should prioritize longitudinal studies to gain a better understanding of the development and progression of orthosomnia, as well as the development and testing of interventions aimed at preventing or alleviating its negative impacts.

## Figures and Tables

**Figure 1 brainsci-14-01123-f001:**
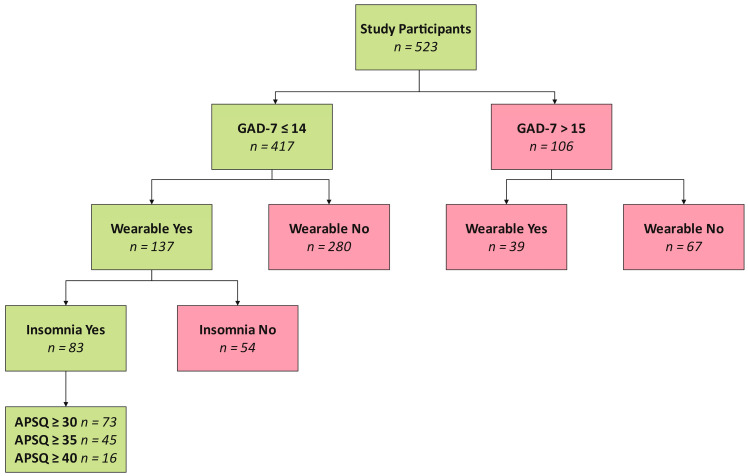
Flowchart of Participant Selection and Criteria Application for Identifying Orthosomnia.

**Table 1 brainsci-14-01123-t001:** Descriptive statistics of the sample (n = 523).

Continuous Variables	Median	IQR	95% CI	α	Minimum–Maximum
Age (in years)	21	5	22.9–24.3	-	18–70
APSQ total	31	11	30.6–32.0	0.91	10–50
AIS	5	5	5.1–5.7	0.83	0–15
GAD-7	9	7	9.3–10.2	0.84	0–21
**Categorical Variables**	**n**	**%**	**95% CI**	**-**	**-**
Wearable	176	34%	34.6–40.1	-	-
AIS					
Insomnia, ≥6	243	46.5%	42.1–50.8		
Healthy, <6	280	53.5%	49.2–57.9		
GAD-7					
Severe anxiety (15–21)	106	20.3%	16.9–23.9		
Moderate anxiety (10–14)	145	27.7%	23.9–31.8		
Mild anxiety (5–9)	181	34.6%	30.5–38.9		
Minimal anxiety (0–4)	91	17.4%	14.3–20.9		
Orthosomnia cases (APSQ ≥ 45) *	6	1.1%	0.25–2.0		
Orthosomnia cases (APSQ ≥ 40) *	16	3.0%	1.6–4.5		
Orthosomnia cases (APSQ ≥ 35) *	45	8.6%	6.2–11.0		
Orthosomnia cases (APSQ ≥ 30) *	73	14.0%	10.9–16.9		
Orthosomnia cases (APSQ ≥ 25) *	76	14.5%	11.5–17.6		
Orthosomnia cases (APSQ ≥ 20) *	82	15.7%	12.6–18.8		
Orthosomnia cases (APSQ ≥ 15) *	82	15.7%	12.6–18.8		

Notes: IQR = Interquartile range, APSQ = Anxiety and Preoccupation about Sleep Questionnaire, AIS = Athens Insomnia Scale, GAD-7 = Generalized Anxiety Disorder 7-item scale, Wearable = Wearable technology ownership, referring to commercial electronic devices worn on the body to track health metrics, Orthosomnia = anxiety and obsession with getting perfect sleep, often exacerbated by the use of sleep-tracking devices. * The APSQ score ranges from 10 (minimum) to 50 (maximum), with higher scores indicating more severe symptoms. To define orthosomnia cases, we used multiple cumulative cutoffs. For example, an APSQ cutoff of ≥15 includes scores from 10 to 15, while a cutoff of ≥20 includes scores from 10 to 20 and so on. This cumulative approach allows for a more nuanced classification of orthosomnia severity across different score ranges.

**Table 2 brainsci-14-01123-t002:** Distribution of orthosomnia by sex at various Anxiety and Preoccupation about Sleep Questionnaire (APSQ) cutoffs (n = 523).

Measure	APSQ ≥ 30	APSQ ≥ 35	APSQ ≥ 40
Total cases/total (prevalence)	73/523 (14.0%)	45/523 (8.6%)	16/523 (3.1%)
Female cases/total	60/422 (14.2%)	40/422 (9.5%)	14/422 (3.3%)
Male cases/total	13/101 (12.9%)	5/101 (5.0%)	2/101 (2.0%)
χ^2^ (df = 1)	0.123	2.121	0.491
*p*-Value	0.726	0.145	0.483
Cramer’s V	0.0153	0.0637	0.0307
Odds ratio [95% CI]	1.12 [0.59, 2.13]	2.01 [0.77, 5.23]	1.70 [0.38, 7.60]
Difference in proportions [95% CI]	0.0135 [−0.0599, 0.0868]	0.0453 [−0.00542, 0.0960]	0.0134 [−0.0187, 0.0455]
Gamma [95% CI]	0.0575 [−0.263, 0.378]	0.336 [−0.0886, 0.760]	0.259 [−0.440, 0.958]
Gamma standard error	0.164	0.216	0.356
Kendall’s Tau-b	0.0153	0.0637	0.0307
Kendall’s Tau-b t	0.350	1.46	0.700
Kendall’s Tau-b *p*	0.726	0.145	0.484
Mantel–Haenszel χ^2^ (df = 1)	0.123	2.121	0.491
Mantel–Haenszel *p*	0.726	0.145	0.484

Notes: Percentages are rounded to one decimal place. CI = Confidence Interval. The *p*-values are based on the χ^2^ test unless otherwise specified. Mantel–Haenszel test is used for trend analysis. APSQ = Anxiety and Preoccupation about Sleep Questionnaire.

**Table 3 brainsci-14-01123-t003:** Results of the age comparison between the case and non-case groups at various Anxiety and Preoccupation about Sleep Questionnaire (APSQ) cutoffs (n = 523).

Group	APSQ ≥ 30	APSQ ≥ 35	APSQ ≥ 40
n (case/non-case)	73/450	45/478	16/507
Case median age (in years) (IQR)	20 (5)	21 (5)	19 (5)
Non-case median age (in years) (IQR)	21 (5)	21 (5)	21 (5)
Mann–Whitney U	14,469	10,130	2937
U *p*-value	0.1	0.516	0.06
Rank biserial correlation	0.119	0.0581	0.276

Notes: U = Mann‒Whitney U statistic; APSQ = Anxiety and Preoccupation about Sleep Questionnaire.

**Table 4 brainsci-14-01123-t004:** Results of the insomnia comparison based on Athens Insomnia Scale (AIS) between the case and non-case groups at various Anxiety and Preoccupation about Sleep Questionnaire (APSQ) cutoffs (n = 523).

Group	APSQ ≥ 30	APSQ ≥ 35	APSQ ≥ 40
n (case/non-case)	73/450	45/478	16/507
Case median AIS (IQR)	8 (2)	8 (3)	9 (5)
Non-case median AIS (IQR)	4 (4)	4 (4)	4 (3)
Mann–Whitney U	6459	4030	1400
U *p*-value	<0.001	<0.001	<0.001
Rank biserial correlation	0.607	0.625	0.655

Notes: AIS = Athens Insomnia Scale. APSQ = Anxiety and Preoccupation about Sleep Questionnaire. All tests use the alternative hypothesis Ha: μ Case ≠ μ Non-Case. Rank biserial correlation is provided as an effect size measure for the Mann–Whitney U test.

## Data Availability

The data that support the findings of this study are available from the corresponding author (Haitham Jahrami) upon request. The data are not publicly available due to ethical restrictions.

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
