# Peer review of "Prevalence of Orthosomnia in a General Population Sample: A Cross-Sectional Study"

_brainsci, 2024, doi:10.3390/brainsci14111123_

Round 1
Reviewer 1 Report
Comments and Suggestions for Authors
Title: Prevalence of Orthosomnia in a General Population Sample: A Cross-Sectional Study
Summary: Orthosomnia is an unhealthy preoccupation with achieving quality sleep as determined by sleep trackers. The prevalence of orthosomnia in the general population is hypothesized to be 3-5%. But the exact prevalence and potential risk factors associated with high prevalence is unknown. The authors aim to quantify the prevalence of orthosomnia and evaluate the potential risk factors. This study addresses a knowledge gap critical to inform detection and clinical management of individuals presenting with insomnia. To do so, the authors apply a novel algorithm to define orthosomnia.
General Concept Comments:
1. [Page 5 Lines 188-199] These lines describe identification of orthosomnia using the algorithm, but the logic of total N included at application of each algorithmic criteria is missing. Please present this as a flow chart with number of individuals at each criterion. For example, 176 patients used a sleep tracker. Of these 176 who meet the initial criteria, for how many was the GAD score <15 and >=15? Next, among the N with GAD score <15, how many has AIS score >= 6. And among those who satisfied the above criteria, how many had the different APSQ score ranges?
2. It is unclear what population the different scores regions at the APSQ score capture. For example, APSQ >= 45 and APSQ >=40, does the APSQ >=40 includes all individuals from 40-50? Or is it referring to the range from 40-44? The current cut offs used is confusing. Would recommend using a defined range, as applicable.
3. [Table 1] Average age of the population, AIS score and the GAD-7, all appear to be skewed. So, would recommend using the median, inter-quartile range and total range instead of mean (SD)
4. The study is powered sufficiently to calculate the prevalence of orthosomnia in general, and describe association with risk factors such as age, sex etc. But I suspect the study is underpowered to look at the difference by sex within each of the orthosomnia categories as defined by the APSQ score cutoffs. This is evident from the wide range of the 95% confidence intervals for the calculated odds ratio in Section 3.3. (APSQ >30: line 220; APSQ>35: line 224; APSQ >40: line 228). This is a limitation of the study and needs to addressed in the discussion section of the paper.
5. Another major concern with the presentation and interpretation of the data is the inaccurate definition of a non-cases (of orthosomnia) in analysis for Table 3 & 4. As the very initial step in defining orthosomnia in the algorithm is “sleep tracker usage”, non-cases should ideally be individuals who use sleep tracker but do not meet the other criteria (GAD>15, AIS <6 or APSQ <30). But Tables 3 & 4, capture non-cases as all individuals not meeting the criteria for insomnia including those not using a sleep tracker.
Other issues:
1. Have the authors considered evaluating familial history of insomnia or anxiety as a covariate?
2. There is a lot of redundancy in the way information is presented in the paper. This impacts the flow of the paper. I would recommend the authors keep an eye out for repetition across the sections and present them more concisely. To highlight a few,
- [Page 2 Lines 92-93] The age range of the population is repeated in the consecutive sentences.
- [Page 2 Lines 93-94] The sentence states “The sample was mostly female, with 81% (n= 422) of the participants being female, and 19% (n=101) were male”. In the absence of missing data on sex, 81% female automatically indicates the inverse (19%) is male. The current statement is redundant. I would recommend simply stating “Majority of the participants were female (81%).”
- [Page 3 Line 112] There seem to be a typo “has” included. Please verify the same.
- [Pages 5-6] There is much redundancy in the information presented in Sections 3.2 and 3.3. Section 3.3 for the most part has all the information described in Section 3.2. Please combine the two sections to avoid repetition.
3. [Page 5 Lines 193-195] It is unclear what the authors are referring to by “APSQ total had to be (45-5 multiple cut off testes) or higher.” Please clarify.
4. [Page 7 Line 247] The authors suggest a trend towards younger age in case at higher cutoffs. But a slight increase in the age of cases is noted at APSQ cutoff >35, from APSQ cutoff >=30. Please rephrase the interpretation of the data.
5. [Page 7 Lines 254-256] The authors present results from both parametric (student t-test) and non-parametric (mann-whitney). I would recommend the authors evaluate the distribution of AIS score, and based on that present either one of the tests, as applicable.
6. [Page 8 Lines 299-300] The authors attribute the likelihood that individuals are more likely to use sleep tracking devices, but this is out of the scope of this study and is not represented by the cited paper “Delayed sleep wake phase disorder in adolescents: an updated review”.
Author Response
Reviewer 1
Title: Prevalence of Orthosomnia in a General Population Sample: A Cross-Sectional Study
Summary: Orthosomnia is an unhealthy preoccupation with achieving quality sleep as determined by sleep trackers. The prevalence of orthosomnia in the general population is hypothesized to be 3-5%. But the exact prevalence and potential risk factors associated with high prevalence is unknown. The authors aim to quantify the prevalence of orthosomnia and evaluate the potential risk factors. This study addresses a knowledge gap critical to inform detection and clinical management of individuals presenting with insomnia. To do so, the authors apply a novel algorithm to define orthosomnia.
Dear Reviewer 1, Thank you for taking the time to review our manuscript and provide valuable suggestions. We greatly appreciate your input, and we have carefully considered each of your recommendations. During the revision, we have carefully considered your suggestions to enhance the quality of the manuscript. Below, we address each of your proposed comments in point-by-point fashion.
General Concept Comments:
- [Page 5 Lines 188-199] These lines describe identification of orthosomnia using the algorithm, but the logic of total N included at application of each algorithmic criteria is missing. Please present this as a flow chart with number of individuals at each criterion. For example, 176 patients used a sleep tracker. Of these 176 who meet the initial criteria, for how many was the GAD score <15 and >=15? Next, among the N with GAD score <15, how many has AIS score >= 6. And among those who satisfied the above criteria, how many had the different APSQ score ranges?
Authors’ response: Thank you so much for this suggestion, we found the idea of the flowchart very appealing. Thus, we created a flowchart and provided an explanation of it as follows: “To illustrate the methodology our algorithm employs in identifying orthosomnia, Figure 1 presents a flowchart that delineates the number of individuals meeting each specified criterion. Our study began with 523 participants, initially categorized based on their Generalized Anxiety Disorder (GAD-7) scores. Among these participants, 417 scored 14 or lower on the GAD-7 scale. Within this subset, 137 participants used a wearable sleep tracker, from which we identified 83 individuals diagnosed with insomnia symptoms. Further stratification based on the Anxiety and Preoccupation about Sleep Questionnaire scores revealed the following: 73 participants scored 30 or higher, 45 scored 35 or higher, and 16 scored 40 or higher. We classified individuals meeting all these criteria as "Cases" of orthosomnia.”
- It is unclear what population the different scores regions at the APSQ score capture. For example, APSQ >= 45 and APSQ >=40, does the APSQ >=40 includes all individuals from 40-50? Or is it referring to the range from 40-44? The current cut offs used is confusing. Would recommend using a defined range, as applicable.
Authors’ response: We provided a detailed clarification about the cutoffs when presenting results (Table 1) as follows: “*The APSQ score ranges from 10 (minimum) to 50 (maximum), with higher scores indicating more severe symptoms. To define orthosomnia cases, we used multiple cumulative cutoffs. For example, an APSQ cutoff of ≥15 includes scores from 10 to 15, while a cutoff of ≥20 includes scores from 10 to 20 and so on. This cumulative approach allows for a more nuanced classification of orthosomnia severity across different score ranges.”
- [Table 1] Average age of the population, AIS score and the GAD-7, all appear to be skewed. So, would recommend using the median, inter-quartile range and total range instead of mean (SD).
Authors’ response: Agreed, we updated the method to the following: “Since the data did not follow a normal distribution, we reported the median and inter-quartile range (IQR) for continuous variables, and frequencies and percentages for categorical variables.”
In Table 1 and subsequent results/tables, we reported Median and IQR.
- The study is powered sufficiently to calculate the prevalence of orthosomnia in general, and describe association with risk factors such as age, sex etc. But I suspect the study is underpowered to look at the difference by sex within each of the orthosomnia categories as defined by the APSQ score cutoffs. This is evident from the wide range of the 95% confidence intervals for the calculated odds ratio in Section 3.3. (APSQ >30: line 220; APSQ>35: line 224; APSQ >40: line 228). This is a limitation of the study and needs to addressed in the discussion section of the paper.
Authors’ response: Agreed, we added the following limitation: “A significant limitation of this study is the sex imbalance in our sample, with women making up about 80% of the participants. This overrepresentation of women could skew the results and limit the generalizability of our findings to the broader population. Although our analysis did not show significant differences in orthosomnia prevalence between males and females, the small number of male participants (n=101) compared to females (n=422) reduces the statistical power to detect any existing differences. This imbalance may also mask potential sex-specific patterns or risk factors associated with orthosomnia. Future studies should aim for a more balanced sex representation to provide a more accurate picture of orthosomnia prevalence and characteristics across sexes.”
- Another major concern with the presentation and interpretation of the data is the inaccurate definition of a non-cases (of orthosomnia) in analysis for Table 3 & 4. As the very initial step in defining orthosomnia in the algorithm is “sleep tracker usage”, non-cases should ideally be individuals who use sleep tracker but do not meet the other criteria (GAD>15, AIS <6 or APSQ <30). But Tables 3 & 4, capture non-cases as all individuals not meeting the criteria for insomnia including those not using a sleep tracker.
Authors’ response: We appreciate this valuable suggestion. We would like to highlight that the two solutions provide different depths of data.
The reviewer’s proposed solution defines non-cases as individuals who use a sleep tracker but do not meet other criteria (GAD > 15, AIS < 6, or APSQ < 30/35/40) and is based on a “matched” case-control analysis. Our initial solution aligns better with the nature of the data and design, specifically a cross-sectional approach. Since the two approaches are complementary and yield similar conclusions, we have added the proposed analyses alongside the initial solutions.
In the methods we explained:
“We classified those who did not meet one or more of these criteria as "Non-Cases."
For sensitivity analyses, we also re-defined "Non-Cases" as individuals who scored 14 or lower on the GAD-7 scale, owned a sleep tracker, had an AIS score of less than 6, or an APSQ score of less than 30, providing an alternative definition of non-cases.”
In the results we provided update as follows:
“When we redefined "Non-Cases" as individuals who scored 14 or lower on the GAD-7 scale, owned a sleep tracker, had an AIS score of less than 6, or an APSQ score of less than 30, 35, and 40, we observed no significant differences compared to the full sample. The results from the independent samples t-tests indicated that for the APSQ ≥ 30 cutoff, the Mann-Whitney U statistic was 314.50 with a p-value of 0.480, yielding a rank biserial correlation of 0.14. For this cutoff, the median age was 20.00 for both the "No" group (N = 10) and the "Yes" group (N = 73). Similarly, at the APSQ ≥ 35 cutoff, the Mann-Whitney U statistic was 734.00 with a p-value of 0.266 and a rank biserial correlation of 0.14, with median ages of 20.00 for the "No" group (N = 38) and 21.00 for the "Yes" group (N = 45). Finally, for the APSQ ≥ 40 cutoff, the Mann-Whitney U statistic was 437.50, p = 0.254, and rank biserial correlation was 0.18, with median ages of 21.00 for the "No" group (N = 67) and 19.00 for the "Yes" group (N = 16). These analyses indicate that the redefined criteria for "Non-Cases" did not yield significant differences in age across the various APSQ cutoffs.”
We also explained the following: “When we reexamined the insomnia, comparison based on the Athens Insomnia Scale (AIS) at various APSQ cutoffs, using the redefined criteria for "Non-Cases" mentioned above. Our analysis revealed significant differences between the case and non-case groups. For the APSQ ≥ 30 cutoff, we calculated a Mann-Whitney U statistic of 222.50 with a p-value of 0.043, resulting in a rank biserial correlation of 0.39. The median AIS score for the "No" group (N = 10) was 7.00, while for the "Yes" group (N = 73), it was 8.00. At the APSQ ≥ 35 cutoff, the Mann-Whitney U statistic was 572.00, with a p-value of 0.009 and a rank biserial correlation of 0.33. The median AIS score for the "No" group (N = 38) remained at 7.00, and for the "Yes" group (N = 45), it was also 8.00. Finally, for the APSQ ≥ 40 cutoff, we found a Mann-Whitney U statistic of 344.50, with a p-value of 0.025 and a rank biserial correlation of 0.36. The median AIS score for the "No" group (N = 67) was 8.00, while for the "Yes" group (N = 16), it was 9.00. These results indicate significant differences in AIS scores between the case and non-case groups at all APSQ cutoffs, highlighting the severity of insomnia symptoms in the case group.”
Other issues:
- Have the authors considered evaluating familial history of insomnia or anxiety as a covariate?
Authors’ response: We did not include familial history of insomnia or anxiety as a covariate in this study. Thus, we suggested them for future research as follows: “In this study, we excluded familial history of insomnia or anxiety as a covariate in our analyses. However, we recognize that these factors can significantly influence the development and manifestation of insomnia symptoms. Future research should evaluate familial history as a covariate to better understand its impact on insomnia and anxiety outcomes.”
- There is a lot of redundancy in the way information is presented in the paper. This impacts the flow of the paper. I would recommend the authors keep an eye out for repetition across the sections and present them more concisely. To highlight a few,
- [Page 2 Lines 92-93] The age range of the population is repeated in the consecutive sentences.
Authors’ response: We removed age range of the population as follows: “The study sample included 523 adults aged 18--70 years, with a median = 21 years, IQR = 5. Majority of the participants were female (81%). The majority of participants were single (83%, n = 435), and 17% (n = 88) were married.”
- [Page 2 Lines 93-94] The sentence states “The sample was mostly female, with 81% (n= 422) of the participants being female, and 19% (n=101) were male”. In the absence of missing data on sex, 81% female automatically indicates the inverse (19%) is male. The current statement is redundant. I would recommend simply stating “Majority of the participants were female (81%).”
Authors’ response: We simplified sex presentation as follows: “The study sample included 523 adults aged 18--70 years, with a median = 21 years, IQR = 5. Majority of the participants were female (81%). The majority of participants were single (83%, n = 435), and 17% (n = 88) were married.”
- [Page 3 Line 112] There seem to be a typo “has” included. Please verify the same.
Authors’ response: “has” has been remove.
- [Pages 5-6] There is much redundancy in the information presented in Sections 3.2 and 3.3. Section 3.3 for the most part has all the information described in Section 3.2. Please combine the two sections to avoid repetition.
Authors’ response: Agree, we have combined Sections 3.2 and 3.3 as follows: “3.2 Prevalence Rates and Characteristics of Orthosomnia
Our classification algorithm indicates that the prevalence of orthosomnia varies depending on the ASPQ score threshold. Using a conservative cut-off of ≥40, we classified 16 participants (3.0%, 95% CI 1.6% - 4.5%) as having orthosomnia. At a moderate cut-off of ≥35, this number increased to 45 participants (8.6%, 95% CI 6.2% - 11.0%), and with a lenient cut-off of ≥30 or higher, 73 participants (14.0%, 95% CI 10.9% - 16.9%) were identified. See Table 1. Further increases in leniency beyond the ≥30 ASPQ cut-off yielded minimal changes in classification: 76 participants (14.5%, 95% CI 11.5% - 17.6%) met the criteria at a cut-off of ≥25, and 82 participants (15.7%, 95% CI 12.6% - 18.8%) were classified at ≥20. Conversely, only 6 individuals (1.1%, 95% CI 0.25% - 2.0%) met the criteria using a cut-off of ≥45. See Table 1.
At an ASPQ cut-off of >30, the overall prevalence of orthosomnia reached 14.0% (73/523), with 14.2% (60/422) among females and 12.9% (13/101) among males. Statistical analysis using the χ² test revealed no significant difference between sexes (χ² = 0.123, p = 0.726, Cramer's V = 0.0153), with an odds ratio of 1.12 [95% CI: 0.59, 2.13] and a difference in proportions of 0.0135 [95% CI -0.0599, 0.0868]. As the ASPQ cut-off increased to >35, the overall prevalence dropped to 8.6% (45/523), with females at 9.5% (40/422) and males at 5.0% (5/101). This reduction also showed no statistical significance (χ² = 2.121, p = 0.145, Cramer's V = 0.0637), featuring an odds ratio of 2.01 [95% CI 0.77, 5.23] and a difference in proportions of 0.0453 [95% CI -0.00542, 0.0960]. At the highest cut-off of >40, the prevalence further decreased to 3.1% (16/523), with 3.3% (14/422) of females and 2.0% (2/101) of males affected. Again, this difference was not statistically significant (χ² = 0.491, p = 0.483, Cramer's V = 0.0307), with an odds ratio of 1.70 [95% CI: 0.38, 7.60] and a difference in proportions of 0.0134 [95% CI -0.0187, 0.0455]. See Table 2.”
- [Page 5 Lines 193-195] It is unclear what the authors are referring to by “APSQ total had to be (45-5 multiple cut off testes) or higher.” Please clarify.
Authors’ response: We provided a detailed clarification about the cutoffs when presenting results (Table 1) as follows: “*The APSQ score ranges from 10 (minimum) to 50 (maximum), with higher scores indicating more severe symptoms. To define orthosomnia cases, we used multiple cumulative cutoffs. For example, an APSQ cutoff of ≥15 includes scores from 10 to 15, while a cutoff of ≥20 includes scores from 10 to 20 and so on. This cumulative approach allows for a more nuanced classification of orthosomnia severity across different score ranges.”
- [Page 7 Line 247] The authors suggest a trend towards younger age in case at higher cutoffs. But a slight increase in the age of cases is noted at APSQ cutoff >35, from APSQ cutoff >=30. Please rephrase the interpretation of the data.
Authors’ response: Agreed, we rephrased the sentence as follows: “Cases tended to be younger than non-cases at APSQ ≥ 30 and APSQ ≥ 40, but not APSQ ≥ 35, differences were not statistically significant.”
- [Page 7 Lines 254-256] The authors present results from both parametric (student t-test) and non-parametric (mann-whitney). I would recommend the authors evaluate the distribution of AIS score, and based on that present either one of the tests, as applicable.
Authors’ response: In the revised manuscript, we have reported only the non-parametric Mann-Whitney U test for the comparison of scores. This decision was made after evaluating the distribution of the data, which indicated that the data did not meet the assumptions necessary for parametric testing. We appreciate your suggestion and have ensured that our analysis is consistent with the distribution characteristics of the data.
- [Page 8 Lines 299-300] The authors attribute the likelihood that individuals are more likely to use sleep tracking devices, but this is out of the scope of this study and is not represented by the cited paper “Delayed sleep wake phase disorder in adolescents: an updated review”.
Authors’ response: We deleted the above-mentioned paragraph from the discussion.

Reviewer 2 Report
Comments and Suggestions for Authors
Comments:
Abstract: "Individuals with orthosomnia were significantly younger (mean age of 19 years) than those without (mean age of 21 years (p = 0.058)". This result does not show statistical significance, so the null hypothesis should be accepted.
Abstract: The conclusions are very strong for a cross-sectional study. The type of study cannot infer an estimate for the population as a whole, but rather for a respective point in time in the specific population from which the sample was established.
Introduction: The hypothesis is overestimated, and the type of study addressed does not provide robust answers to the hypothesis raised. Therefore, the hypothesis should be focused on the specific population and not in general. It would be appreciable to adjust the study hypotheses.
Material and methods: The instruments for behavioral assessment of sleep and anxiety phenotype had an effective traceability capacity.
Material and methods: It would be appreciable to include the effect of the test for the nonparametric Mann‒Whitney comparison. Furthermore, the generalized linear model was used to predict a possible association, since the assumptions for more robust tests such as regression were not met. Therefore, an alternative is the GLM.
The manuscript has scientific merit and addresses a topic of importance to health. I can give a positive opinion after the checklist and verification of the suggestions.
Author Response
Reviewer 2
Dear Reviewer 2, Thank you for taking the time to review our manuscript and provide valuable suggestions. We greatly appreciate your input, and we have carefully considered each of your recommendations. During the revision, we have carefully considered your suggestions to enhance the quality of the manuscript. Below, we address each of your proposed comments in point-by-point fashion.
Abstract: "Individuals with orthosomnia were significantly younger (mean age of 19 years) than those without (mean age of 21 years (p = 0.058)". This result does not show statistical significance, so the null hypothesis should be accepted.
Authors’ response: We agreed with the reviewer, and we changed the statement to: “Individuals with orthosomnia were not significantly different in terms of age and sex.”
Abstract: The conclusions are very strong for a cross-sectional study. The type of study cannot infer an estimate for the population as a whole, but rather for a respective point in time in the specific population from which the sample was established.
Authors’ response: We agreed with the reviewer; we toned down the language as follows: “This study offers initial insights into the prevalence of orthosomnia within our sample at a specific time. The findings reveal notable rates of orthosomnia among individuals using sleep-tracking devices; however, we must acknowledge the limitations inherent in a cross-sectional design.”
Introduction: The hypothesis is overestimated, and the type of study addressed does not provide robust answers to the hypothesis raised. Therefore, the hypothesis should be focused on the specific population and not in general. It would be appreciable to adjust the study hypotheses.
Authors’ response: We agreed with the reviewer; we toned down the language as follows: On the basis of our clinical observations, we hypothesized that the prevalence of orthosomnia in our sample is approximately 3-5% [6-8].”
Material and methods: The instruments for behavioral assessment of sleep and anxiety phenotype had an effective traceability capacity.
Thank you for your feedback. We appreciate your acknowledgment of the effective traceability capacity of the instruments used for assessing sleep and anxiety phenotypes. We ensured that these tools were validated and reliable to provide accurate and meaningful data for our study.
For further accountability/transparency, in the present research we reported reliability using Cronbach's alpha . In the paper, we reported the following: “
In methods: We evaluated the reliability of the measures used in the present study using Cronbach's alpha. This statistical measure assesses the internal consistency of the scales, ensuring that the items within each scale reliably measure the same construct.
In results: The scales used in this study yielded the following Cronbach's alpha values: the GAD-7 had a reliability coefficient of 0.89, the APSQ showed a coefficient of 0.91, and the AIS recorded a coefficient of 0.88. These values indicate that each measure possesses high internal consistency.”
Material and methods: It would be appreciable to include the effect of the test for the nonparametric Mann‒Whitney comparison. Furthermore, the generalized linear model was used to predict a possible association, since the assumptions for more robust tests such as regression were not met. Therefore, an alternative is the GLM.
Authors’ response: We agreed with the reviewer, on the two analytical comments:
Effect Size for Mann-Whitney U Test: We have now highlighted the rank biserial correlation as the effect size for the Mann-Whitney U comparisons. We added to the methods:
We used rank-biserial correlation to assess the effect size of the Mann–Whitney U test.
Generalized Linear Model (GLM): We agree that using a GLM can be a robust alternative for predicting associations when assumptions for regression are not met.
We have incorporated a GLM analysis to explore potential associations, ensuring a comprehensive evaluation of the data. We reported the following:
In methods: We also employed the generalized linear model (GLM) to analyze the relationship between orthosomnia status and predictor variables, including age and sex. Researchers chose logistic regression due to the binary nature of the dependent variable, orthosomnia cutoff at 40, and applied a logit link function. The model assumes a binomial distribution to account for the dichotomous outcome and evaluates model fit and complexity using metrics such as R-squared, AIC, and BIC.
In results: The model yielded an R-squared value of 0.02, indicating that the predictors explained a modest proportion of the variance. Age had a non-significant effect (p = 0.142) with an odds ratio of 1.08. Similarly, sex did not significantly impact the outcome (p = 0.491), showing an odds ratio of 1.69. The model converged successfully; the predictors did not significantly contribute to explaining the outcome variable.
The manuscript has scientific merit and addresses a topic of importance to health. I can give a positive opinion after the checklist and verification of the suggestions.
Authors’ response: Thank you for your valuable feedback and for endorsing our work. We have addressed all the comments you provided, as outlined above.

Reviewer 3 Report
Comments and Suggestions for Authors
This paper estimates the prevalence of orthosomnia in the general population through a survey of 523 participants and examines its relationship with the use of wearable sleep-tracking devices. The organization and content of the paper are generally well-structured, but there are several issues that need to be addressed before acceptance:
1. The introduction is too brief and lacks a discussion of relevant current research, as well as a thorough explanation of the problem this study aims to solve.
2. With women comprising 81% of the sample, could this skew the results? This should be addressed in the discussion section.
3. How was the effectiveness of the classification algorithm validated? Are its results consistent with expert judgments? Is there any specific data provided to support this?
4. Why wasn't an analysis of the reliability and validity of the questionnaires included?
5. What are the contributions of this study? The authors need to elaborate on this in the introduction.
Author Response
Reviewer 3
This paper estimates the prevalence of orthosomnia in the general population through a survey of 523 participants and examines its relationship with the use of wearable sleep-tracking devices. The organization and content of the paper are generally well-structured, but there are several issues that need to be addressed before acceptance:
Dear Reviewer 3, Thank you for taking the time to review our manuscript and provide valuable suggestions. We greatly appreciate your input, and we have carefully considered each of your recommendations. During the revision, we have carefully considered your suggestions to enhance the quality of the manuscript. Below, we address each of your proposed comments in a point-by-point fashion.
- The introduction is too brief and lacks a discussion of relevant current research, as well as a thorough explanation of the problem this study aims to solve.
Authors’ response: We agreed with the reviewer; we expanded the introduction with a major focus on the explanation of the problem this study aims to solve, as follows:
Optimal sleep is essential for physical and mental health, wellbeing, and a good quality of life [1]. However, many individuals struggle to attain a restful night's sleep [9]. In recent years, interest in monitoring and enhancing sleep quality has surged, partly driven by the increasing accessibility and popularity of sleep tracking devices worldwide [2]. This trend has given rise to a new phenomenon called "orthosomnia", a term coined by Baron et al. in 2017 [3]. Orthosomnia refers to an unhealthy preoccupation with achieving perfect sleep as defined by these tracking devices [3, 4].
The growing prevalence of sleep tracking technology has revolutionized how individuals engage with their sleep patterns [10]. Wearable devices and smartphone applications now offer users detailed metrics on sleep duration, quality, and cycles [10]. While these tools provide valuable insights, they may also heighten anxiety and obsession over sleep data for some users [3, 4]. This preoccupation can paradoxically lead to sleep disturbances, creating a cycle of anxiety and poor sleep [5].
Recent research has begun to explore the implications of widespread sleep tracking [2]. Robbins et al. conducted a systematic review of studies using commercial sleep tracking technology, highlighting both the potential benefits and limitations of these devices [2]. Robbins team found that while sleep trackers can increase user engagement with sleep health, concerns exist about their accuracy and the potential for increased anxiety in some users [2]. Glazer Baron et al. further examined how consumer sleep technology data are utilized in behavioral sleep medicine interventions, underscoring the growing integration of these technologies in clinical practice [4].
The concept of orthosomnia has gained increasing attention in sleep medicine literature. Baron et al.'s original case series [3] described patients whose sleep issues worsened due to strict adherence to sleep tracker data, which often conflicted with objective sleep measures [3]. This phenomenon raises important questions about the psychological impact of quantifying sleep and the potential for technology induced sleep anxiety [3].
Despite growing clinical awareness of orthosomnia in sleep medicine, we still lack a clear understanding of its prevalence and characteristics in the general population [6]. Although case studies and anecdotal reports have highlighted the negative impact of orthosomnia on sleep quality and daytime functioning [3, 5], previous studies have not estimated its prevalence in the general population [6]. This absence of epidemiological data limits our understanding of the issue's extent and its potential public health implications [8].
Moreover, the relationship between orthosomnia and demographic factors such as age and sex remain unexplored [6]. Identifying these potential risk factors could provide valuable insights for targeted prevention and intervention strategies. Moreover, we need to investigate the association between orthosomnia and insomnia symptoms, as this may inform differential diagnosis and treatment approaches in sleep medicine.
The present cross-sectional study aims to address these knowledge gaps by quantitatively estimating the prevalence of orthosomnia in a general population sample. We developed and applied a novel classification algorithm that combines several validated scales for insomnia, anxiety and sleep related concerns with information about wearable device usage. As secondary objectives, we examined age and sex as potential risk factors associated with orthosomnia and explored the relationship between orthosomnia and insomnia symptoms.
Our findings may contribute to the growing body of knowledge on the intersection of technology and sleep health. By providing the first quantitative estimates of orthosomnia prevalence and associated factors, we aim to inform healthcare providers, sleep medicine specialists, and public health officials about this emerging concern. Furthermore, this research may guide future investigations into the long-term impacts of sleep tracking on sleep perception and behavior, ultimately leading to more effective strategies for promoting healthy sleep in the digital age.
- With women comprising 81% of the sample, could this skew the results? This should be addressed in the discussion section.
Authors’ response: We agreed with the reviewer; we added the following limitation:
A significant limitation of this study is the sex imbalance in our sample, with women making up about 80% of the participants. This overrepresentation of women could skew the results and limit the generalizability of our findings to the broader population. Although our analysis did not show significant differences in orthosomnia prevalence between males and females, the small number of male participants (n=101) compared to females (n=422) reduces the statistical power to detect any existing differences. This imbalance may also mask potential sex-specific patterns or risk factors associated with orthosomnia. Future studies should aim for a more balanced sex representation to provide a more accurate picture of orthosomnia prevalence and characteristics across sexes.
- How was the effectiveness of the classification algorithm validated? Are its results consistent with expert judgments? Is there any specific data provided to support this?
Authors’ response: We acknowledged that our reported prevalence rates of orthosomnia are based on algorithmic computations and have not been formally validated. We highlighted this more explicitly in our discussion as follows:
The lack of established diagnostic criteria or validated scales for orthosomnia presents a major challenge in validating any classification method for this condition. We developed our algorithm based on current literature and expert clinical judgment, incorporating elements that align with the theoretical conceptualization of orthosomnia. Although the prevalence rate meets clinical expectations, we recognize that this alone does not validate the algorithm. We did not directly compare our algorithm's results to expert clinical judgments on a case-by-case basis, nor did we provide specific validation data. Therefore, we should view our classification algorithm as a preliminary attempt to operationalize orthosomnia for research purposes, rather than a definitive diagnostic tool. Future research should focus on developing and validating a specific scale for orthosomnia, comparing our algorithm's results with expert clinical diagnoses, and refining the algorithm based on emerging research and expert consensus. We appreciate the reviewer for highlighting this important issue and will explicitly discuss these limitations in future revisions of the manuscript.
- Why wasn't an analysis of the reliability and validity of the questionnaires included?
Authors’ response: In response to concerns about the reliability and validity of our measures, we would like to clarify that the scales used in this study have been previously validated in earlier research. However, in the present research, we reported reliability using Cronbach. In the paper, we reported the following:
In methods: We evaluated the reliability of the measures used in the present study using Cronbach's alpha. This statistical measure assesses the internal consistency of the scales, ensuring that the items within each scale reliably measure the same construct.
In results: The scales used in this study yielded the following Cronbach's alpha values: the GAD-7 had a reliability coefficient of 0.89, the APSQ showed a coefficient of 0.91, and the AIS recorded a coefficient of 0.88. These values indicate that each measure possesses high internal consistency.
- What are the contributions of this study? The authors need to elaborate on this in the introduction.
Authors’ response: We thank the reviewer for this comment; we expanded the introduction and focus on the contributions of this study, as follows: “Our findings may contribute to the growing body of knowledge on the intersection of technology and sleep health. By providing the first quantitative estimates of orthosomnia prevalence and associated factors, we aim to inform healthcare providers, sleep medicine specialists, and public health officials about this emerging concern. Furthermore, this research may guide future investigations into the long-term impacts of sleep tracking on sleep perception and behavior, ultimately leading to more effective strategies for promoting healthy sleep in the digital age.”

Round 2
Reviewer 1 Report
Comments and Suggestions for Authors
Please clarify the following description in the text:
- [Page 5 Lines 193-195] It is unclear what the authors are referring to by “APSQ total had to
be (45-5 multiple cut off testes) or higher.” Please clarify. It is still unclear what the authors are referring to by "(45-5 multiple cut off testes) or higher."

Author Response
Reviewer 1
Thank you for your positive feedback and valuable input. Your guidance and input have been invaluable in refining our work, and we are grateful for your support.
[Page 5 Lines 193-195] It is unclear what the authors are referring to by “APSQ total had to be (45-5 multiple cut off tested) or higher.” Please clarify. It is still unclear what the authors are referring to by "(45-5 multiple cut off testes) or higher."
Authors’ response: Thank you for your re-review and your suggestion.
In the manuscript we explained that: “To determine the best threshold for the ASPQ total score to be used in the definition of orthosomnia, we tested various cutoffs ranging from 15 to 45. The APSQ score ranges from a minimum of 10 to a maximum of 50, with higher scores indicating more severe symptoms and no specified cutoff. In this research, we employed multiple cumulative cutoffs. Specifically, for an APSQ score of ≥45, the range corresponds to scores from 10 to 45; for ≥40, the range is 10 to 40; for ≥35, it includes scores from 10 to 35; for ≥30, the range is 10 to 30; for ≥25, it covers scores from 10 to 25; for ≥20, the range is 10 to 20; and finally, for ≥15, it includes scores from 10 to 15. We stopped at the cutoff of 15 because it became clear that it was saturated; specifically, the number of orthosomnia cases defined by an APSQ score of ≥20 was equal to those defined by a score of ≥15 (i.e., Orthosomnia cases APSQ ≥ 20 = Orthosomnia cases APSQ ≥ 15).”
Please Refer to Results in Table 1.
Table 1. Descriptive statistics of the sample (N = 523).
Continuous variables |
Median |
IQR |
95% CI |
α |
Minimum - Maximum |
Age (in years) |
21 |
5 |
22.9 - 24.3 |
- |
18 - 70 |
APSQ Total |
31 |
11 |
30.6 – 32.0 |
0.91 |
10 - 50 |
AIS |
5 |
5 |
5.1 - 5.7 |
0.83 |
0 - 15 |
GAD-7 |
9 |
7 |
9.3 - 10.2 |
0.84 |
0 - 21 |
Categorical variables |
N |
% |
95% CI |
- |
- |
Wearable |
176 |
34 % |
34.6 - 40.1 |
- |
- |
AIS Insomnia, ≥ 6 Healthy, < 6 |
243 280 |
46.5 % 53.5 % |
42.1 – 50.8 49.2 – 57.9 |
|
|
GAD-7 Severe anxiety (15 - 21) Moderate anxiety (10 - 14) Mild anxiety (5 - 9) Minimal anxiety (0 - 4) |
106 145 181 91 |
20.3 % 27.7 % 34.6 % 17.4 % |
16.9 – 23.9 23.9 – 31.8 30.5 – 38.9 14.3 – 20.9 |
|
|
Orthosomnia cases (APSQ ≥ 45)* |
6 |
1.1 % |
0.25 – 2.0 |
|
|
Orthosomnia cases (APSQ ≥ 40)* |
16 |
3.0 % |
1.6 – 4.5 |
|
|
Orthosomnia cases (APSQ ≥ 35)* |
45 |
8.6 % |
6.2 – 11.0 |
|
|
Orthosomnia cases (APSQ ≥ 30)* |
73 |
14.0 % |
10.9 – 16.9 |
|
|
Orthosomnia cases (APSQ ≥ 25)* |
76 |
14.5 % |
11.5 – 17.6 |
|
|
Orthosomnia cases (APSQ ≥ 20)* |
82 |
15.7 % |
12.6 – 18.8 |
|
|
Orthosomnia cases (APSQ ≥ 15)* |
82 |
15.7 % |
12.6 – 18.8 |
|
|
Notes: IQR = Interquartile range, APSQ = Anxiety and Preoccupation with Sleep Questionnaire, AIS = Athens Insomnia Scale, GAD-7 = Generalized Anxiety Disorder 7-item scale, Wearable = Wearable technology ownership, referring to commercial electronic devices worn on the body to track health metrics, Orthosomnia = anxiety and obsession with getting perfect sleep, often exacerbated by the use of sleep-tracking devices. *The APSQ score ranges from 10 (minimum) to 50 (maximum), with higher scores indicating more severe symptoms. To define orthosomnia cases, we used multiple cumulative cutoffs. For example, an APSQ cutoff of ≥15 includes scores from 10 to 15, while a cutoff of ≥20 includes scores from 10 to 20 and so on. This cumulative approach allows for a more nuanced classification of orthosomnia severity across different score ranges.

Reviewer 3 Report
Comments and Suggestions for Authors
I have a major concern regarding this article. The authors mentioned in their response that the gender imbalance could lead to biases in the results, which may overturn the existing conclusions. Regarding this issue, I think that simply elaborating on it in the limitations section is not sufficient. The authors need to provide specific statistical results to actually illustrate this situation, that is, whether it has an impact and the extent of the impact.
Author Response
Reviewer 3
Thank you for your positive feedback and valuable input. Your guidance and input have been invaluable in refining our work, and we are grateful for your support.
I have a major concern regarding this article. The authors mentioned in their response that the gender imbalance could lead to biases in the results, which may overturn the existing conclusions. Regarding this issue, I think that simply elaborating on it in the limitations section is not sufficient. The authors need to provide specific statistical results to actually illustrate this situation, that is, whether it has an impact and the extent of the impact.
Authors’ response: Thank you for your re-review and your suggestion.
In the manuscript we explained that:
In methods: “To address the sex imbalance, we repeated the chi-square test by matching for age and wearable status, comparing 101 males and 101 females to ensure a balanced and reliable analysis of the association between orthosomnia status and sex”.
In results: “In the matched analysis (for age and wearable status) comparing 101 males and 101 females, a statistically significant result was obtained for the APSQ cutoff of 30 (p = 0.02); however, no statistically significant results were obtained at the APSQ cutoffs ≥ 35 and ≥ 40, both p > 0.05.”
In discussion: “In the matched analysis comparing equal numbers of males and females, we found a statistically significant difference in orthosomnia status at the APSQ cutoff of 30. However, at the higher cutoffs of 35 and 40, there were no statistically significant differences. This suggests that sex differences may be more apparent at lower APSQ cutoff levels.”

Round 3
Reviewer 3 Report
Comments and Suggestions for Authors
accept